

# The effect of mountaineering on the grit of college students: an empirical study

Lun Li, ZuWang Chu, FuLin Li, JiaoJiao Li, Kang Wang and Yun Zhou

China University of Geosciences (Wuhan), Wuhan, China

## ABSTRACT

**Objective.** Although ample evidence in the literature suggests a correlation between general sports participation and resilience, information on the potential impact of specific sports activities on resilience is lacking. Therefore, we attempt to evaluate the effect of mountaineering on the grit of college students.

**Methods.** The study recruited 12 healthy college students for a three-day mountaineering activity. Then, the grit scores of the students before and after mountaineering were tested using the Chinese version of the original grit scale (Grit-O). During the mountaineering process, the oxygen saturation of the subjects was measured and recorded using a portable finger clipper. The scores of the Lake Louise Scale (2018 Lake Louise Acute Mountain Sickness Score, LLS) were used to distinguish whether the mountaineers had acute mountain sickness (AMS). Independent $t$-tests and paired $t$-tests were performed on the data separately.

**Results.** A significant correlation exists between the total score of grit among college students before and after mountaineering ($r = 0.646$, Sig $< 0.05$). However, the total score did not significantly improve after mountaineering ($p = 0.054$), the effort scores of all college students increased significantly before and after mountaineering ($p = 0.045$). A significant correlation is also observed between the total score of grit among college students who have not suffered from AMS ($r = 0.764$, Sig $< 0.05$). However, no significant improvement occurs ($p = 0.075$). Meanwhile, no significant correlation exists between the efforts ($r = 0.499$, Sig $> 0.05$) and interests ($r = 0.562$, Sig $> 0.05$) of college students before and after mountaineering. AMS has no significant correlation with the resilience of college students before and after mountaineering, in terms of effort ($r = 0.456$, Sig $> 0.05$) and interest ($r = 0.601$, Sig $> 0.05$), while no significant difference was observed between the total resilience score, effort and interest of mountaineering and non-mountaineering college students before and after mountaineering (all $p > 0.05$).

**Conclusion.** In the short term, mountaineering has a certain enhancing effect on cultivating the grit of college students. However, the existing grit scale cannot fully reflect the resilience scores of mountaineers before and after. Hence, other situational dimensions should be added to the design of the grit scale.

Corresponding author
Yun Zhou, zhouyun16307@163.com

## INTRODUCTION

The concept of grit proposed by Duckworth, which pertains to a highly important and predictable noncognitive personality trait for success and performance, has received
widespread attention in the past decade (*Duckworth et al., 2007*; *Duckworth, Quinn & Tsukayama, 2009*; *Duckworth, 2011*; *Wei et al., 2014*; *Wang, 2016*). Studies have shown that grit predicts not only the severity and duration of physical activity (*Reed, Pritschet & Cutton, 2013*; *Ueno, Satoshi & Atsushi, 2018*), but also the level of physical health (*Cosgrove, Chen & Castelli, 2018*). This concept, developed in a specific context with some debate surrounding its creation, has limited application across diverse measurement contexts, including sports psychology (*Reed, Pritschet & Cutton, 2013*; *Martin et al., 2015*; *Larkin, O'Connor & Williams, 2016*; *Tedesqui & Young, 2017*; *Tedesqui & Young, 2018*; *Gupta & Sudhesh, 2019*). The lack of broader testing raises concerns about its generalizability, reliability, and validity. Research efforts investigating the concept as a domain-specific or domain-general construct offer valuable insights into its applicability. With the rise and development of positive psychology, a growing number of researchers have focused on cultivating individual positive psychological qualities, rather than simply preventing the occurrence of psychological disorders. However, few scholars have studied the cultivation of grit as a trait in positive psychology.

The original grit scale (Grit-O) consists of 12 items and uses a five-point Likert scale. The scale consists of two dimensions, namely, consistency of interests and persistence in effort (*Duckworth et al., 2007*). *Duckworth & Quinn (2009)* revised Grit-O by deleting four items, thus resulting in the formation of the Short Grit Scale (Grit-S) with eight items. Confirmatory factor analysis showed good fit for each factor. *Nan (2018)* simultaneously revised the Chinese version of Grit-O and Grit-S, and conducted reliability and validity tests on the college student population. The results indicate that the structural validity of Grit-O is good among college students, and is superior to the testing effect of Grit-S. Therefore, compared with Grit-S, Grit-O is more suitable for measuring the resilience quality of Chinese university students, thus providing a reliable measurement tool for studying the grit level of university students.

In very high altitude areas, when the degree of hypoxia exceeds an individual's adaptive regulation range, it can cause a series of pathological and physiological reactions in the body, leading to the occurrence of high-altitude diseases (*Hackett & Roach, 2001*). The function of the cardiovascular system is to transport the oxygen required for tissue metabolism and make adaptive adjustments based on changes in the amount of oxygen required for tissue metabolism (*Hooper & Mellor, 2011*). This study considers not only the convenience and speed of on-site testing, but also the ability to reflect the cardiovascular system's adaptive regulation to hypoxia, using blood oxygen saturation as a measurement indicator.

There are studies suggesting that rising to high altitudes (>2,500 m) may lead to acute mountain sickness, which typically manifests as non-specific symptoms such as headache, dizziness, loss of appetite, nausea, vomiting, insomnia, and fatigue (*Basnyat & Murdoch, 2003*; *Rodway, Hoffman & Sanders, 2003*; *Gallagher & Hackett, 2004*). These symptoms may pose additional difficulties for climbers during the climbing process (*Campbell et al., 2015*).

Common knowledge implies that individuals engaging in outdoor sports and especially in regular and extreme mountaineering are exceptionally healthy and hardened (*Habelt et al., 2023*). However, no scholar has conducted research on the effect of mountaineering on

**Table 1 Details of participants.**

| Total | Female | Age (Years) | Height (cm) | Weight (kg) | Experienced more than 5,000 m | Drug-free | Having been to high altitude in recent 3 months |
|---|---|---|---|---|---|---|---|
| 12 | 4 | $27 \pm 9$ | $173.4 \pm 8.8$ | $67.9 \pm 11.5$ | 6 | 0 | 2 |

grit (*Schimschal et al., 2021*). To our knowledge, this study is the first to conduct research on the grit of mountaineers. It combines the oxygen saturation ($SpO_2$) and acute mountain sickness (AMS) monitoring of mountaineers with the impact of extreme environments on their resilience, and with their grit scale to analyse their grit. This study also attempts to determine whether mountaineering is the best practice for cultivating grit at individual and collective levels.

## METHODS

### Location
We conducted a three-day mountaineering activity from July 17–19, 2023 at the second peak of Mt. Siguniang (5,454 m) in Sichuan Province, China. We tested the grit score of mountaineers using Grit-O in Chengdu (Sichuan, China, altitude 484 m). We used the Lake Louise Scale to test AMS scores at altitudes of 3,200–5,454 m and a portable finger clipper to measure the $SpO_2$ levels of mountaineers. These altitudes are considered very high altitudes.

### Research subjects
Twelve university students from China University of Geosciences, who voluntarily participated in this mountaineering activity, were evaluated. The subjects were not excellent athletes and had not systematically participated in various sports activities; they were all general sports enthusiasts. They are all college students (undergraduates or postgraduates), who enjoy outdoor activities, but they have climbed mountains less than twice. Only healthy, nonsmoking, nonmedicated and unmeasured mountaineers participated in this survey. The average age of the subjects was $27 \pm 9$ years old, and the average weight was $67.9 \pm 11.5$ kilograms. The test met the ethical review of the Ethics Committee of China University of Geosciences (No. CUG2022-04-01), and received written informed consent from all participants (Table 1).

Values for age, height and weight were mean ± SD.

### Sea level measurement
Before mountaineering, all college students underwent physical examinations at designated hospitals. Only healthy individuals were selected for the research. On July 2, the subjects were informed of the test content. On July 3, participants simulated data collection. $SpO_2$ was recorded using a fingertip oximeter (PHILIPS, DB12, Suzhou Erda Medical Equipment Co., Ltd., China).
## Measurement during mountaineering

Before the mountaineering activity, the participants rode a car from Chengdu to Siguniang Mountain Town, Xiaojin County, Aba Tibetan and Qiang Autonomous Prefecture, Sichuan Province (3,200 m) and then walked to the base camp (4,307 m). While it is possible to choose to ride a horse to the base camp, all participants arrived at the base camp on foot.

The two perseverance quality tests before and after mountaineering were completed in Chengdu. All the mountaineers independently completed the offline questionnaire.

The three-day mountaineering activity started with the participants' arrival at Siguniang Mountain Town on the first morning, and ended upon their return to Siguniang Mountain Town on the third evening. The height of the subjects' daily ascent was controlled between 700 m to 1,100 m. Meanwhile, the time for ascent and return did not exceed 16 h. The $SpO_2$ and Lake Louise score were measured three times a day: 30 min after waking in the morning, at the highest point of altitude during the day, and after all public activities at night. During the blood oxygen test, the subjects were all sitting. In the morning and evening, the subjects were measured indoors after resting for 5 min, with the finger clipper placed on their fingertip for more than 15 s. During the day, the subjects moved to the highest point, where they were immediately tested to collect the data by using the finger clipper for 15 s with their finger placed inside their clothing. The Lake Louise score was obtained using on-site questionnaires and recorded methods. Nine sets of mountaineering team data were measured in three days.

Throughout the ascent, we continuously monitored the team members for symptoms of altitude sickness, advising them to rest or descend as necessary to mitigate additional safety risks. The tester followed the subject throughout the entire climbing process, in order to ensure the safety of the climbers and the progress of the climb, such as resting at the campsite or taking breaks during the journey, the tester randomly communicated with the subjects to understand their feelings. After the conversation is over, the tester takes notes.

## Symptom scores

The original grit scale (Grit-O) consists of 12 items and uses a five-point Likert scale (1 = not at all like me; 5 = very much like me). The scale consists of two dimensions, namely, consistency in interests and persistence in effort (*Duckworth et al., 2007*). The items include the ability to persevere in adversity (for example, "New ideas and new projects sometimes distract me from previous ones".), as well as the consistency of long-term efforts (for example, "I have overcome setbacks to conquer an important challenge".). The internal consistency coefficient of the scale is 0.85.

The Lake Louise Scale (LLS) was used for the diagnosis of AMS (*Hackett & Oelz, 1992*; *Roach et al., 2018*). AMS was diagnosed according to the latest altitude increase, the presence of headaches, and at least one of the following symptoms: headache, gastrointestinal (GI) upset, fatigue, dizziness. The symptoms were graded from 0 to 3, with 0 meaning no symptoms at all and 1 to 3 meaning mild, moderate and severe symptoms, respectively. The self-reported LLSs were categorised as mild AMS (3–5 points), moderate AMS (6–9 points) and severe AMS (10–12 points). A total score of 3 or more indicate the presence of AMS. We used LLS scores of 3 or more to diagnose AMS even though climbers at this

threshold might be declared in dangerous situations with no AMS symptoms. At very high altitudes, the consequences of false positives are still minimal.

## Data analysis

Each data table was holistically reviewed by the lead investigator for completeness. Since the sample size in this study is less than 50 for each group, we used the Shapiro–Wilk test to check the normality of the data before applying these parameters. The results showed that the $p$-values for all data groups were greater than 0.05, indicating that all data conform to a normal distribution. The differences in resilience and $SpO_2$ between the acute high-altitude disease group and the nonacute high-altitude disease group were evaluated through independent sample $t$-tests and paired sample $t$-tests, as distinguished by the self-report from Lake Louise. We calculated the correlation between AMS, grit and $SpO_2$ among college climbers at different altitudes. The software used was SPSS 20.0 (IBM, Chicago, IL, USA) with a significance level (alpha) of 0.05 (two-tailed). Values were given as mean $\pm$ SD.

# RESULTS

All mountaineers rode a car to Siguniang Mountain Town on the first day to purchase materials, contact logistics support and handle procedures. The next day, they departed from Siguniang Mountain Town and hiked lightly to the headquarters. On the third day, they set off from the headquarters to climb the second peak of Mount Siguniang and then hiked back to the town. During the three days, a dedicated person set up the camp, cooked and cleaned utensils. All other tasks were completed in collaboration with the college mountaineering team.

## Lake Louise scores of mountaineers

On the first day of mountaineering, two people developed AMS, and one of the climbers scored 6 points on the LLS. During the subsequent mountaineering process, nine climbers experienced acute mountain sickness at varying degrees with the highest score reaching 11 points. Three mountaineers had never experienced AMS (Fig. 1).

## $SpO_2$ of mountaineers

All mountaineers had a decreasing trend in $SpO_2$ during the mountaineering process. After completing the journey, their levels returned to normal, and their $SpO_2$ at the end of the mountaineering were generally lower than those at the beginning of entering the mountain. However, through a paired sample $t$-test, significance was not achieved ($t = 0.693$, $r = -0.275$, Sig $= 0.503$). The highest $SpO_2$ of 96 points appeared at noon on July 17, and the lowest $SpO_2$ of 62 points appeared on the evening of July 18 (Fig. 2).

## Relationship between AMS and $SpO_2$ of mountaineers

Through testing the Lake Louise scores and $SpO_2$ of the mountaineers for three days, it was found that only two of them developed AMS when they arrived at the base camp on the first day. One of the two individuals developed AMS on the first morning because of motion sickness and mild vomiting during the journey to Siguniang Mountain Town by
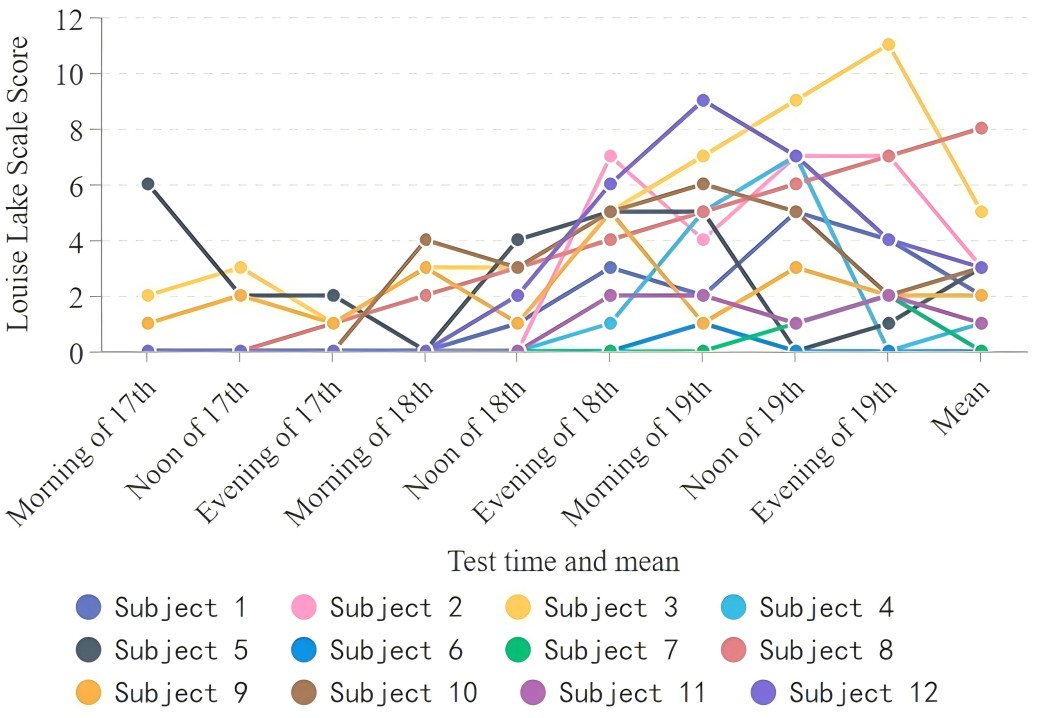

**Figure 1** Statistics of the Lake Louise scores for mountaineers.

car. Therefore, they scored higher on the LLS on the first day. During the mountaineering process on the second and third days, the mountaineers all experienced varying degrees of AMS. Four mountaineers were unable to reach the summit. Throughout the entire mountaineering process, no significant relationship was observed between AMS and $SpO_2$ of mountaineers except for the situation at noon on the first day, which demonstrated a significant correlation (Table 2).

## Comparison of grit scores for mountaineers

Through a statistical analysis of the grit scores of mountaineers before and after climbing, a significant correlation exists between the total score of grit among college students before and after mountaineering ($r = 0.646$, Sig $< 0.05$). However, the total score did not significantly improve after mountaineering ($p = 0.054$). The correlation between the effort scores of all climbers before and after climbing is not significant, but the effort scores of all college students increased significantly before and after mountaineering ($p = 0.045$). There was no significant difference in interest scores among all climbers before and after climbing, and the correlation was not significant.

By calculating the mean of the LLS for mountaineers, the subjects were divided into two groups: those with AMS and those without AMS. An analysis of the resilience scale scores of the two groups of mountaineers revealed a significant correlation between the total resilience scores of mountaineers who did not suffer from AMS before and after mountaineering ($r = 0.764$, Sig $< 0.05$). However, no significant difference was observed.

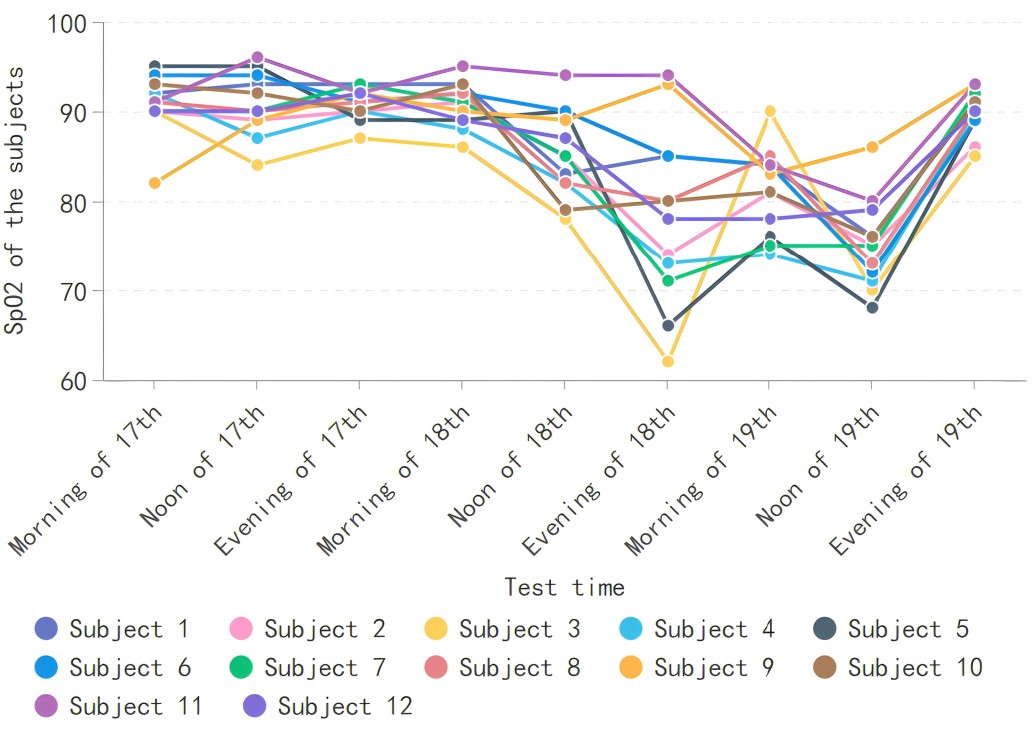

**Figure 2** SpO$_2$ of subjects.

**Table 2** Comparison of SpO$_2$ between groups with and without AMS.

|  |  | AMS | | Non AMS | | P |
|---|---|---|---|---|---|---|
|  |  | n | SpO$_2$(%) | n | SpO$_2$(%) |  |
| 17th | Morning | 1 | 95 | 11 | 90.5 ± 3.1 | 0.192 |
|  | Noon | 1 | 84 | 11 | 91.4 ± 2.8 | 0.032 |
|  | Evening | 0 | – | 12 | 90.8 ± 1.7 | – |
| 18th | Morning | 3 | 89.7 ± 3.5 | 9 | 91.1 ± 2.2 | 0.41 |
|  | Noon | 3 | 82.3 ± 6.7 | 9 | 86.3 ± 4.1 | 0.23 |
|  | Evening | 7 | 76.9 ± 10.7 | 5 | 80.6 ± 9.3 | 0.544 |
| 19th | Morning | 7 | 80.7 ± 5.5 | 5 | 82 ± 3.9 | 0.664 |
|  | Noon | 8 | 75.8 ± 5.1 | 4 | 73.8 ± 5.1 | 0.533 |
|  | Evening | 5 | 88.8 ± 3.1 | 7 | 90.7 ± 1.7 | 0.198 |

Mountaineers with AMS showed no significant correlation in the total grit score, effort score and interest score, while mountaineers without AMS showed no significant correlation in the effort and interest dimensions.

Grouping mountaineers into groups based on whether they reached the summit demonstrated that the mountaineers who reached the summit and those who did not reach the summit improved in terms of overall grit, effort and interest on average. However, no significant correlation was observed between them (Table 3).

**Table 3 Comparison of grit scores of mountaineers.**

| | | Total | AMS | | Summit | |
|---|---|---|---|---|---|---|
| | | | Yes | None | Yes | None |
| | n (Number of female) | 12(4) | 5(2) | 7(2) | 8(2) | 4(2) |
| Grit | Pre-mountaineering | 40.75 ± 5.58 | 41.2 ± 8.04 | 40.431 ± 3.699 | 41.25 ± 4.37 | 39.75 ± 8.22 |
| | Pro-mountaineering | 43.5 ± 4.76 | 44.4 ± 5.32 | 42.86 ± 4.63 | 42.63 ± 4.173 | 45.25 ± 6.02 |
| | correlation coefficient ($r$) | 0.646* | 0.623 | 0.764* | 0.547 | 0.87 |
| | $P$ | 0.054 | 0.319 | 0.0754 | 0.371 | 0.079 |
| Effort | Pre-mountaineering | 21.83 ± 3.61 | 20.8 ± 4.66 | 22.57 ± 2.82 | 22 ± 2.78 | 21.5 ± 5.45 |
| | Pro-mountaineering | 24.08 ± 3.23 | 24.6 ± 2.19 | 23.71 ± 3.95 | 23.5 ± 3.46 | 25.25 ± 2.75 |
| | Correlation coefficient ($r$) | 0.499 | 0.456 | 0.751 | 0.549 | 0.656 |
| | $P$ | 0.452 | 0.109 | 0.291 | 0.203 | 0.172 |
| Interests | Pre-mountaineering | 18.92 ± 4.32 | 20.4 ± 3.58 | 17.86 ± 4.74 | 19.25 ± 4.83 | 18.25 ± 3.594 |
| | Pro-mountaineering | 19.42 ± 2.71 | 19.8 ± 3.42 | 19.14 ± 2.34 | 19.13 ± 2.3 | 20 ± 3.74 |
| | Correlation coefficient ($r$) | 0.562 | 0.601 | 0.573 | 0.499 | 0.892 |
| | $P$ | 0.638 | 0.69 | 0.417 | 0.935 | 0.133 |

**Notes.**
*Sig < 0.05.

# DISCUSSION

We must act with caution when explaining the results of dividing subjects into two groups that change over time. All participants were college students, with an average age difference of nine years and a weight difference of 8.8 kilograms. They climbed and worked together as a team by sharing workload and maintaining the same schedule. The $SpO_2$ and AMS symptoms of each participant were measured using the same testing tools at the same time and altitude. However, considering that the instrument might have been affected by altitude and temperature, our discussion is limited to a general discussion of the data. To avoid data fraud, we also conducted interviews to understand the specific situation of mountaineers regarding acute hyperreflexes and resilience during the mountaineering process.

Perseverance is a psychological quality that reflects an individual's ability to pursue long-term goals persistently for a long period and exhibit high interest or enthusiasm towards these goals (*Duckworth et al., 2007*). Through data analysis, we found that the resilience score of each mountaineer improved before and after climbing. However, only the grit score showed a significant correlation (Sig < 0.05) and nonsignificant improvement. We attempted to analyse whether mountaineering had a positive effect on improving the resilience of the mountaineers by testing their grit before and after mountaineering. We believe that the reason for this situation is that mountaineering has a very clear goal, which is to reach the summit. Therefore, when college students overcome unfavourable conditions, such as environmental and physical decline to achieve the summit goal, the quality of their perseverance improves. This trend is consistent with the research conclusion of *Rutberg et al. (2020)*.

Effort refers to doing something despite difficulty (*Meyer et al., 2021*). Interest refers to the psychological inclination of people to explore certain things or engage in certain activities (*Peng, 2019*). The data from this study show no significant correlation between the effort and interest dimensions of college students before and after mountaineering. During the mountaineering activity, despite the discomfort in the environment and their body, college students persisted in climbing the mountain on the premise of safety. Given that reaching the top of the mountain was an obvious goal of the activity, many of the subjects were able to persist in achieving clear goals, which is a similar trend to the research findings of *Cosgrove, Chen & Castelli (2018)*. According to the conversations of mountaineers, all mountaineers were interested in mountaineering, and the college students who had mountaineering experiences had always been interested in mountaineering, which is consistent with Habert's research (*Habelt et al., 2023*). However, the stability of interest among first-time college students during mountaineering were challenged because of the hardships of the process and the impact of the environment on them, which differs from their initial understanding of mountaineering. Some of the mountaineers expressed that they might no longer participate in mountaineering activities in the future, because the high mountain environment had a significant impact on their bodies.

$SpO_2$ reflects the oxygen carrying capacity of human haemoglobin (the proportion of oxygen carrying haemoglobin molecules to total haemoglobin molecules, often expressed as %) and is also an important biological indicator for human adaptation to low oxygen (*Beall, 2007a*; *Beall, 2007b*). After entering high-altitude areas, the partial pressure of atmospheric oxygen decreases, thus decreasing $SpO_2$. We used the LLS to assess the AMS of mountaineers based on $SpO_2$ to demonstrate that AMS affected their physical fitness, cognition and other activities during mountaineering (*Siqués et al., 2009*; *Wu et al., 2010*; *Modesti et al., 2011*; *Tang et al., 2011*; *Norling et al., 2014*). However, our research results indicate that there is no relationship between AMS and $SpO_2$, which is consistent with the findings of some researchers. A number of researchers have reported that $SpO_2$ is not predictive of AMS, particularly on short 1–3 day ascents (*Wagner, Knott & Fry, 2012*).

The results show that only college students who did not suffer from AMS showed a significant correlation in the grit score of resilience. However, no significant difference was observed. This outcome also indicates that during the mountaineering process, climbers were influenced not only by grit in terms of effort and interest, but also by varying degrees in terms of the environment. Prior studies generated findings that have implications for understanding the neurobiological bases of grit. Drawing from functional magnetic resonance imaging approaches, one of the important regions in the brain that has been linked to grit is the medial prefrontal cortex (*Myers et al., 2016*; *Wang et al., 2017*). Under the influence of adverse conditions such as headaches and fatigue caused by AMS, the college students were able to maintain the consistency of their interest and perseverance within a short period (3 days). As such, situational dimensions must be added to the grit scale (*Suzuki et al., 2015*; *Datu, Yuen & Chen, 2018*), especially in high-altitude mountaineering environments, to test resilience quality. Moreover, we believe that the AMS that the subjects experienced during this activity was not particularly obvious. The duration of the activity was not long. Hence, these difficulties did not significantly affect the clear goal of reaching

the summit. This outcome indicates that one critical way to improve grit is by practicing process-oriented goal setting, rather than outcome-focused goal setting. Grit cannot be provided, and it is not a stagnant trait (*Gray et al., 2023*).

To our knowledge, we are the first to study the impact of mountaineering on cultivating grit among college students. Although no significant correlation emerged between the resilience of college students before and after this mountaineering activity, some team members had severely high reflexes and reached the summit, while others had mildly high reflexes and did not reached the summit. The observation of college students during the mountaineering process and the conversations after climbing reflect that the dimension of effort is the key to whether one climbs the summit.

## CONCLUSION

As an extreme sport, mountaineering has a certain promoting effect on cultivating the grit of college students in the short term, the grit score showed a significant correlation (Sig < 0.05) and nonsignificant improvement. However, Grit-O cannot fully reflect the grit scores of mountaineers before and after the climbing activity. In designing a grit scale for mountaineers, other situational dimensions can be included apart from the dimensions of effort and interest.

## LIMITATIONS

Given the consideration of the risks of mountaineering and the safety of mountaineers, the sample size of the participants in this study was small, and the mountaineering duration was not long. All of these factors might have caused certain inaccuracies in data analysis. Future research should focus on the gender differences between males and females.

## PROSPECT

Future research can combine grit with positive psychological indicators, such as happiness and achievement, in mountaineering environments. This modification can fully reflect the role of mountaineering in cultivating grit among college students. Further research can also seek to determine whether individuals with high grit carry risks during mountaineering.

## ACKNOWLEDGEMENTS

We would like to express our gratitude to the college mountaineers and all those who participated in the testing work who contributed to this research.

### Funding

Financial support was received for the research, authorship, and/or publication of this article. This research was supported by the Open Fund from Key Research Institute of Humanities and Social Sciences in Hubei Province-research Center of University Student

Development and Innovation Education and Hubei Leisure Sports Development Research Center Open Fund Project for 2023. The funders had no role in study design, data collection and analysis, decision to publish, or preparation of the manuscript.

### Grant Disclosures

The following grant information was disclosed by the authors:
Open Fund from Key Research Institute of Humanities and Social Sciences in Hubei Province-research Center of University Student Development and Innovation Education and Hubei Leisure Sports Development Research Center Open Fund Project for 2023.

### Competing Interests

The authors declare there are no competing interests.

### Author Contributions

- Lun Li conceived and designed the experiments, performed the experiments, analyzed the data, authored or reviewed drafts of the article, and approved the final draft.
- ZuWang Chu conceived and designed the experiments, analyzed the data, authored or reviewed drafts of the article, and approved the final draft.
- FuLin Li performed the experiments, prepared figures and/or tables, and approved the final draft.
- JiaoJiao Li performed the experiments, prepared figures and/or tables, and approved the final draft.
- Kang Wang performed the experiments, prepared figures and/or tables, and approved the final draft.
- Yun Zhou conceived and designed the experiments, analyzed the data, authored or reviewed drafts of the article, and approved the final draft.

### Human Ethics

The following information was supplied relating to ethical approvals (i.e., approving body and any reference numbers):
The Ethics Committee of China University of Geosciences (Wuhan) (No. CUG2022-04-01).

### Data Availability

The data is available at figshare: Li, Lun (2024). Test data of Mount Siguniang in 2023.csv. figshare. Dataset. https://doi.org/10.6084/m9.figshare.27411366.v1.

### Supplemental Information

Supplemental information for this article can be found online at http://dx.doi.org/10.7717/peerj.19086#supplemental-information.

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
