# Peer review of "The effect of mountaineering on the grit of college students: an empirical study"

_PeerJ, doi:10.7717/peerj.19086_

## Round 0.1 · original submission · Major Revisions

The manuscript titled "The effect of mountaineering on the grit of college students: an empirical study". Pre- and post-summit courage was examined after a three-day mountain climb. The subject matter is new and intriguing, but a more thorough study design is required.

·

Basic reporting

The manuscript is about " The Effect of Mountaineering on the Grit of College Students: An Empirical Study"
The authors investigated the impact of a three-day mountain climb on participants' courage prior to and after the summit.

However, a few points raised in the article warrant further investigation. One of the most important things for climbers to remember is to seek adequate treatment if they see signs of mountain sickness. If these symptoms are detected, the climber should rest, descent, and refrain from additional effort. However, it appears that these therapies were not used in the climbers investigate. Please explain why participants were still encouraged to raise altitude despite having signs of mountain sickness.

Moreover, In Tables 1 and 3, one of the columns shows the number or percentage of female athletes. However, no mention of gender differences in climbing is made in any of the article's sections.

Table 1 states that "experienced more than 5000 m: 6". Does this indicate that each person had 6 climbs above 5,000 meters, or that the average climb for 12 people was 6 climbs?

Table 3 should be re-designed to be more clarified.

Experimental design

It is difficult to conclude that the participants' grit increased with a 5000-meter climb only in 3 days.
The story did not mention the athletes' acclimatization before to the 5000-meter climb. Because going to elevations exceeding 3000 meters necessitates frequent training, a lack of acclimatization might result in potentially fatal conditions for the athlete.
More than courage, having 12 partners can be beneficial in continuing the climb. Because the participants witness their teammates continuing the climb, they give their all to this exercise, and competitiveness with fellow climbers can be regarded a co-factor bias. In other words, the effect of teamwork should be considered in this research.

The main question in method part is mentioned above.
One of the most important things for climbers to remember is to seek adequate treatment if they see signs of acute mountain sickness (AMS). If these symptoms are detected, the climber should rest, descent, and refrain from additional effort. However, it appears that these therapies were not used in the climbers investigate.

Validity of the findings

The topic under study is novel and interesting, but a more detailed study design is needed.

Additional comments

If a new study on mountain climbers is conducted, it will be necessary for subjects to receive further training in order to climb high peaks.

Reviewer 2 ·

Basic reporting

In the introduction, the concept of grip and the association of this term with physical health and the duration of physical activities are analyzed. The authors give a good description of the two Grit Scale versions (Original Grit Scale/Grit-O/12 items and Short Grit Scale/Grit-S-8 items). The body's reactions to hypoxia and the symptoms characteristic of the human body to activities at high altitudes are presented. The sections of the manuscript are clearly and logically structured, the graphs and tables include the results of the dependent variables analyzed. The research direction is clearly defined. The references used are well associated with the investigated topic.

Experimental design

The mountain location and the time interval/3 days allocated to the study are specified. The instruments for assessing AMS, SpO2 levels and Grit score are also presented. The data of the study participants are summarized in Table 1. The research complied with the requirements recommended by the studies involving human subjects/Declaration of Helsinki and informed consent agreements were obtained. The differentiated scores for the 3 self-reported LLS categories are analyzed in detail. The statistical procedures and software used for the calculation are also mentioned.

Validity of the findings

The results are presented in the two graphs and tables quite well, but this section and the conclusions can be improved. The Discussion section identifies and compares the results of this study with other similar research. The limitations of the study have been identified (very small sample size and relatively short duration of the investigation).

Additional comments

1. Why is the analyzed group so small (too few participants) ?
2. Are the participants performance athletes or are they systematically involved in different physical activities? I think that testing the fitness level (in future studies) could influence the results for the functional parameters and the scores on the applied questionnaires.
3. Table 1: The data are presented for the entire group, but the table shows that there are 4 females and 8 males in the study. It might be useful to present the age, height and weight separately by gender. Also, a presentation of the differences between genders for the analyzed dependent variables would complete the results obtained with important information and details.
4. Did you use the t-tests (for paired samples and independent samples) and the Pearson correlation coefficients in the statistical calculation. Was the application of these parametric tests preceded by the analysis of the normality of the data distribution (Shapiro-Wilk test)?
5. Line 181: However, through a paired sample t-test, significance was not achieved (r= -0.275, Sig= 0.503). I think t is correct, not r/correlation coefficient.
6. Table 3 (Comparison of grit scores of mountaineers): Dependent variables (Grit / Effort / Interests) in the first line and the results in the table are moved/transferred to two or three lines (probably when saving the manuscript as a PDF). The data is difficult to analyze....you can format the table to make the results more clearly visible.
7. Lines 124-126: The SpO2 and Lake Louise score were measured three times a day: 30 minutes after waking in the morning, at the highest point of altitude during the day, and after all public activities at night. The differences for these 3 successive measurements could have been highlighted by applying ANOVA with repeated measures. You can perhaps present them in other studies.
8. Table 2 (Comparison of SpO2 between groups with and without AMS). Calculating Effect Size/ Cohen's d would have provided an improvement in the quality of the statistical analysis.
9. The conclusions could better summarize the main directions and results associated with the investigated parameters.

---

## Round 0.2 · accepted · Accept

This revised version is suitable for publication in PeerJ.

Reviewer 2 ·

Basic reporting

Everything is ok in this section.

Experimental design

Everything is ok in this section.

Validity of the findings

Everything is ok in this section.

Additional comments

.The authors responded favorably to all suggestions submitted for the first evaluation of the manuscript.